# Evidence for immune activation in pathogenesis of the HLA class II associated disease, podoconiosis

Mikias Negash [1,2,3] ✉, Menberework Chanyalew[2], Tigist Girma[2], Fekadu Alemu [2], Diana Alcantara[1], Ben Towler [4], Gail Davey[1,5], Rosemary J. Boyton [6], Daniel M. Altmann [7], Rawleigh Howe[2] & Melanie J. Newport [1]

Available evidences suggest that podoconiosis is triggered by long term exposure of bare feet to volcanic red clay soil particles. Previous genome-wide studies in Ethiopia showed association between the HLA class II region and disease susceptibility. However, functional relationships between the soil trigger, immunogenetic risk factors and the immunological basis of the disease are uncharted. Therefore, we aimed to characterise the immune profile and gene expression of podoconiosis patients relative to endemic healthy controls. Peripheral blood immunophenotyping of T cells indicated podoconiosis patients had significantly higher CD4 and CD8 T cell surface HLA-DR expression compared to healthy controls while CD62L expression was significantly lower. The levels of the activation markers CD40 and CD86 were significantly higher on monocytes and dendritic cell subsets in patients compared to the controls. RNA sequencing gene expression data indicated higher transcript levels for activation, scavenger receptors, and apoptosis markers while levels were lower for histones, T cell receptors, variable, and constant immunoglobulin chain in podoconiosis patients compared to healthy controls. Our finding provides evidence that podoconiosis is associated with high levels of immune activation and inflammation with over-expression of genes within the pro-inflammatory axis. This offers further support to a working hypothesis of podoconiosis as soil particle-driven, HLA-associated disease of immuno-pathogenic aetiology.

Podoconiosis is a form of lymphoedema that causes progressive painful swelling of the legs[1]. It is a neglected tropical disease that affects poor communities living in remote highland regions in endemic countries. Available evidence suggests that inflammation in podoconiosis is triggered by an unidentified volcanic clay soil component causing lymphatic fibrosis that leads to swelling and nodular changes in the foot and lower leg[2]. The nature of the soil trigger is not precisely known but epidemiological and geological

[1]Brighton and Sussex Centre for Global Health Research, Department of Global Health and Infection, Brighton and Sussex Medical School, Brighton, UK. [2]Armauer Hansen Research Institute, Addis Ababa, Ethiopia. [3]Department of Medical Laboratory Science, College of Health Sciences, Addis Ababa University, Addis Ababa, Ethiopia. [4]Department of Biochemistry and Biomedicine, School of Life Sciences, University of Sussex, Brighton, UK. [5]School of Public Health, Addis Ababa University, Addis Ababa, Ethiopia. [6]Department of Infectious Disease, Imperial College London, London, UK. [7]Department of Immunology and Inflammation, Imperial College London, London, UK. ✉e-mail: mikiasn2@gmail.com

studies suggest certain climatic conditions are required that contribute for the formation of the soil particles. These include annual rainfall of above 1500 mm, an altitude above 1500 m and surface temperature of 19–21 °C[3]. Such topographic features are abundant globally, but the lack of affordable protective footwear to prevent contact with the soil (and hence disease) limits the geographical distribution of podoconiosis to low-income countries in Africa, Asia and Central and South America[2].

It is estimated that there are about four million cases of podoconiosis globally and the highest prevalence of the disease is observed in Cameroon and Ethiopia with prevalence of 8.08% and 7.45%, respectively[4]. A recent study in 2017 used epidemiological data and modeling techniques to estimate that around 1.5 million cases of podoconiosis in Ethiopia and the prevalence of podoconiosis in the area where the current study was conducted (Gojam Zone, Amhara region) is 3.5%[5,6].

Affected people experience progressive swelling and debilitating pain associated with intermittent episodes of acute adenolymphangitis. These episodes hamper normal day-to-day and agricultural activities which patients' livelihoods depend on, leading to further impoverishment and negative psychosocial impacts[6,7]. Familial clustering of the disease suggests genetic factors play a role in the pathogenesis of the disease. Segregation analysis of multigenerational podoconiosis-affected families in Wolaita, southern Ethiopia, showed an estimated sibling recurrence risk ratio (λs) and heritability of 5.07 and 0.63, respectively[8]. These findings led to the undertaking of genome-wide association study (GWAS) in the Wolaita population[9] and a second larger GWAS in three ethnic populations in Ethiopia[10]. Both these studies reported that variation in HLA class II genes (*DRB1*, *DQA1*, and *DQB1*) was significantly associated with susceptibility to podoconiosis. The Occam's razor explanation for HLA class II disease associations is generally a central role of CD4 T cells.

Taken together with the epidemiological observations, these findings suggest that a mineral or other exogenous compound present in soil triggers an HLA-mediated immune response in susceptible individuals that targets the lymphatic system. Earlier studies by Dr Ernest Price suggested minerals absorbed through the skin are taken up by macrophages and transported to lymph nodes to initiate an inflammatory response[11]. Price also observed that silicate particles caused subendothelial edema, endolymphangitis, collagenisation and obliteration of the lymphatic lumen[12]. More recent studies demonstrated patients had thickened dermal collagen, reduced elastic fibers, dilated and often sclerotic blood vessels, with a moderate lymphoplasmacytic infiltrate which also contained mast cells, and scattered macrophages, but few neutrophils or eosinophils[13].

Although class II HLA molecules typically present "foreign" antigens of pathogen origin, they can also be involved in T-cell-mediated autoimmune or hypersensitivity disease through direct recognition of self-antigens, modified epitopes, or through molecular mimicry[14]. An example of possible relevance to podoconiosis is the pathogenesis of berylliosis in which *HLA-DP* gene products are implicated, either through presentation of a self-peptide modified by beryllium, or through direct binding of beryllium to the HLA-peptide binding groove, thus triggering an inflammatory CD4 T-cell response[15]. Soil from podoconiosis-endemic regions is rich in a number of elements including beryllium, one of six elements shown to be statistically linked to podoconiosis[16]. Beyond the genetic association with the class II HLA region, little is known about the immune response in podoconiosis, reflecting the neglected status of the condition and the communities it affects. However, understanding the mechanisms of disease could accelerate the development of effective treatments and early diagnostic or screening tests to identify susceptible individuals that could help eliminate the condition. As the soil trigger and an experiment model on podoconiosis is lacking, this study aimed to characterize innate and adaptive immunity in podoconiosis through immunophenotyping and RNA expression studies, moving toward improved understanding of the pathogenesis of podoconiosis.

## Results

### Study subjects, clinical, and socio-demography data

A total of 64 podoconiosis patients and 49 healthy controls were enrolled in the study. The majority of the patients (59 of 64, 92.2%) were in stage 2 of the disease clinically based on the Tekola staging system[17], had bilateral disease (58, 90.6%), were males (36, 55.7%), and their mean age was 47.8 ± 12 years. The majority of healthy controls were males (26 of 49, 53.1%), their mean age was 34.4 ± 8 years, and almost all the participants were farmers (Supplementary Table 1).

### Immunophenotyping of peripheral blood mononuclear cells

**T cells.** Surface expression of markers of activation (HLA-DR, CD38), memory (CD62L) and proliferation (Ki-67) was analyzed on CD4 and CD8 T cells from 56 podoconiosis cases and 44 healthy control subjects using a gating strategy as defined in Supplementary Fig. 2. The average percentage of CD4 and CD8 T cells expressing HLA-DR was significantly higher in podoconiosis patients compared to healthy controls ($P < 0.001$ with median of 10.7% vs 7.1% for HLA-DR on CD4 and 23.4% vs 15.8% for HLA-DR expression on CD8 cells, respectively). In contrast, the expression of CD62L on these T-cell subsets was significantly lower in podoconiosis patients compared to healthy controls (CD4CD62L, $P < 0.0001$, with median value of 53.6% vs 63.1%, respectively; CD8CD62L, $P = 0.001$, with median value of 22.8% vs 36.8%, respectively). There were no differences between patients and healthy controls in CD38 and Ki-67 expression on either CD4 or CD8 T-cell subsets (Fig. 1).

Furthermore, we confirmed the increase in activation marker by comparing median MFI of each activation marker among markers ungated. For example, the MFI of HLA-DR on CD4 T cells was 131 in podoconsiosis patients versus 87 in healthy controls ($P < 0.0001$). Similarly, the MFI of HLA-DR on CD8 T cells was significantly different (348 versus 242, respectively, $P = 0.0004$). We observed the same concordance comparing CD62L in the study groups, CD62L MFI on CD4 T cells was 475 and 1539 in patients and controls, respectively ($P < 0.001$); CD62L MFI on CD8 T Cells was significantly lower in patients compared to controls (98.6 versus 187, respectively, $P = 0.0019$).

**Monocytes.** The expression of HLA-DR, CD40, CD86, and CD36 was analyzed on monocytes from 43 podoconiosis patients and 34 healthy controls. Monocytes were first gated into classical, intermediate and non-classical monocyte subsets based on CD14 and CD16 expression (see Supplementary Fig. 3). There were no statistically significant differences in the distribution of the three monocyte subsets between podoconiosis patients and healthy controls (Supplementary Fig. 4).

Expression of the activation markers CD40 and CD86 was significantly higher on classical monocytes from podoconiosis patients compared to healthy controls, with median values of 35.6% vs 25.5% for CD40 and 13.7% vs 7.4% for CD86 ($P = 0.03$, $P = 0.001$, respectively). There were no differences between the two groups in expression of CD40 or CD86 on non-classical and intermediate monocyte subsets, although there was a trend toward higher expression in podoconiosis patients. CD36 expression in the intermediate monocyte subpopulation was significantly higher in podoconiosis patients than in healthy controls with median value of 56.7% vs 33.2% ($P < 0.0001$), as shown in Fig. 2.

**Dendritic cells.** Expression of HLA-DR, CD40 and CD86 on dendritic cells (DCs) was analyzed in 43 podoconiosis patients and 34 healthy controls after they were sorted into the three DC subsets, myeloid (mDC), plasmacytoid (pDC) and cross-presenting (cp-DC) based on their CD11c, CD123 and CD141 expression respectively (see Supplementary Graph 5).

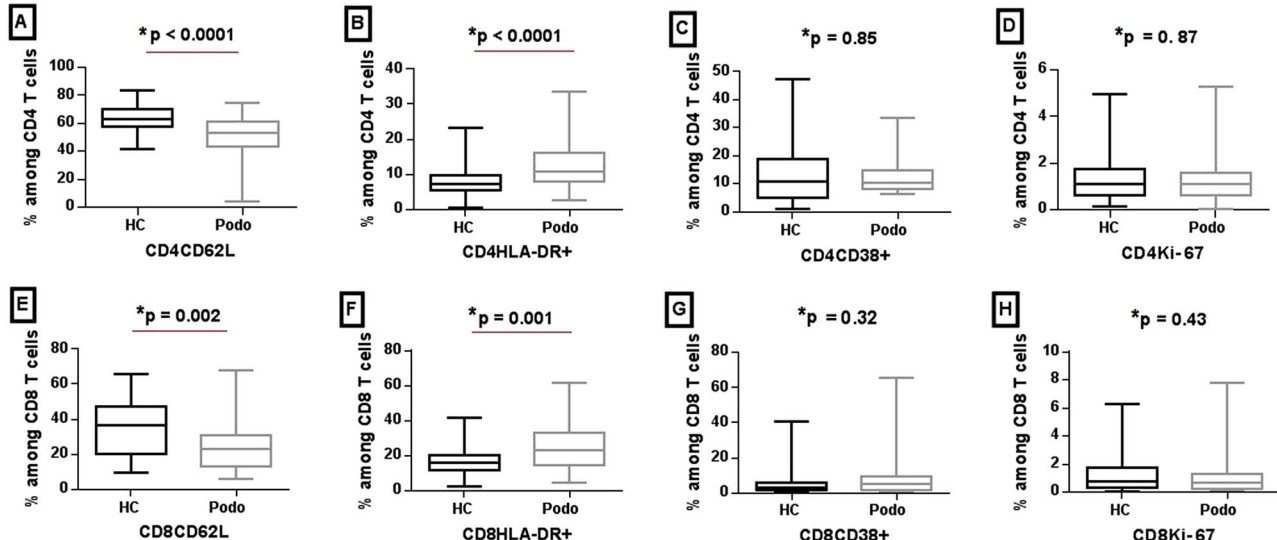

**Fig. 1 | Median expression of CD62L, HLA-DR, CD38, and Ki-67 markers on CD4 and CD8 T cells from podoconiosis patients and healthy controls.** The figure shows box plots depicting median, interquartile range, minimum and maximum values of subset frequencies positive for CD62L, HLA-DR, CD38, and Ki-67 on CD4 (A–D) and CD8 (E–H) T cells, respectively, in 56 podoconiosis patients (Podo) and 44 healthy controls (HC). *P values were derived using the Mann–Whitney U test of two-sided independent t test. Source data that is used to generate this graph is provided as a "Source Data" file Figs. 1–3.

There was no significant difference in the distribution of the three DC subsets between podoconiosis patients and healthy controls (Fig. 3A–C). Evaluation of the activation biomarkers within the three subsets showed that the expression of CD40 was significantly higher in all three DC subsets, in particular on mDCs, in podoconiosis patients compared to healthy controls (median value of 8.4% vs 3.7%, respectively, $P = 0.003$). There was no statistically significant difference between podoconiosis patients and healthy controls in the expression of CD86 on any of the DC subsets, although median levels of CD86 among mDC were higher in podoconiosis patients (median value of 53.7% vs 34.1%, $P = 0.2$) (Fig. 3D–I).

### Transcriptomics analysis of differentially expressed genes using next-generation RNA sequencing

Twenty-four samples matched for PBMC count, RNA yield, sex, and age were selected from each study group to minimize background differences in the transcriptome analysis (see Supplementary Table 2 for summary). Of a total of 24 samples in each group, 23 samples from the healthy controls and 21 samples from podoconiosis patients were pooled into two panels for two sequencing runs (the remaining samples were excluded due to poor library yields). Various quality control (QC) steps were undertaken as briefly described in "Methods".

Only samples with a read count of more than 5 million were considered for downstream pipeline analysis, resulting in 15 healthy controls and 19 podoconiosis cases being included in the differential expression analysis. The analysis returned 242 genes that were differentially expressed between podoconiosis patients and healthy controls with a more than 1.5-fold change and $P < 0.05$. Of these differentially expressed genes, 108 were significantly upregulated, and 134 were significantly downregulated in podoconiosis patients (Fig. 4). Correlation was observed between certain upregulated genes which are involved in antigen processing, inflammation and scavenging of oxidized lipids (*CD80, CD86, HLA-DQB1, CD1A, MSR1*, and *MPO*) and the peripheral blood immunophenotypic findings described above.

The pathway and functional enrichment analysis of differentially expressed genes using DAVID identified more than 15 different clusters with significantly enriched pathways of which the top three were the histones, cell division and DNA/telomere organization. Additional pathways with significant enrichment scores included: cell membrane,

immune response, immunoglobulin domain, and TCR alpha and beta domains. The clusters with the highest enrichment score are presented in Fig. 5A, B for the down- and upregulated GO categories separately.

### Protein–protein interaction network

Enrichment categories with a score of more than 1.3 and a $P$ value of <0.05 were further explored using protein–protein interaction network analysis. This analysis showed that the upregulated genes were mainly involved in lipid metabolism and scavenging of oxidized lipids. This pathway also has known interactions with macrophage and collagen proteins such as CD68 and COL4A2, respectively (Fig. 5D). The protein interaction network for the main downregulated genes was dominated by histones, immunoglobulin lambda variable and constant regions and TCR alpha and beta receptors. A network based on the histones domain is presented below in Fig. 5C.

Lists of representative up- and downregulated genes in podoconiosis for the significantly enriched GO categories are presented in Tables 1 and 2 below, respectively.

### Discussion

Podoconiosis is a noncommunicable disease caused by exposure to an environmental factor on the background of genetic susceptibility associated with variation in *HLA-DR* and *HLA-DQ* genotypes[9,10]. Little is known about the environmental trigger or the pathogenic immune response it induces in podoconiosis. We report here the a detailed phenotypic characterization of T cells, monocytes and DCs in podoconiosis as well as global gene expression data from unstimulated PBMCs comparing podoconiosis patients with healthy controls. Our results suggest that the pathology in podoconiosis is driven by a state of on-going immune activation and inflammation involving key immune response cell types. We have presented evidence of activation of T cells in podoconiosis patients compared to healthy controls, with higher levels of HLA-DR expression on both CD4 and CD8 T-cell subsets while CD62L expression was lower in podoconiosis patients. Similarly, the expression of activation markers like CD40 and CD86 was significantly higher among myeloid DCs and classical monocytes suggesting persistent activation of the cells. The RNA-Seq data demonstrated upregulation of genes involved in antigen processing and presentation (*CD80, CD86, HLA-DQB1*), inflammation and scavenging of metabolically altered lipids (*CD1A, MSR1, OXLR1*) in

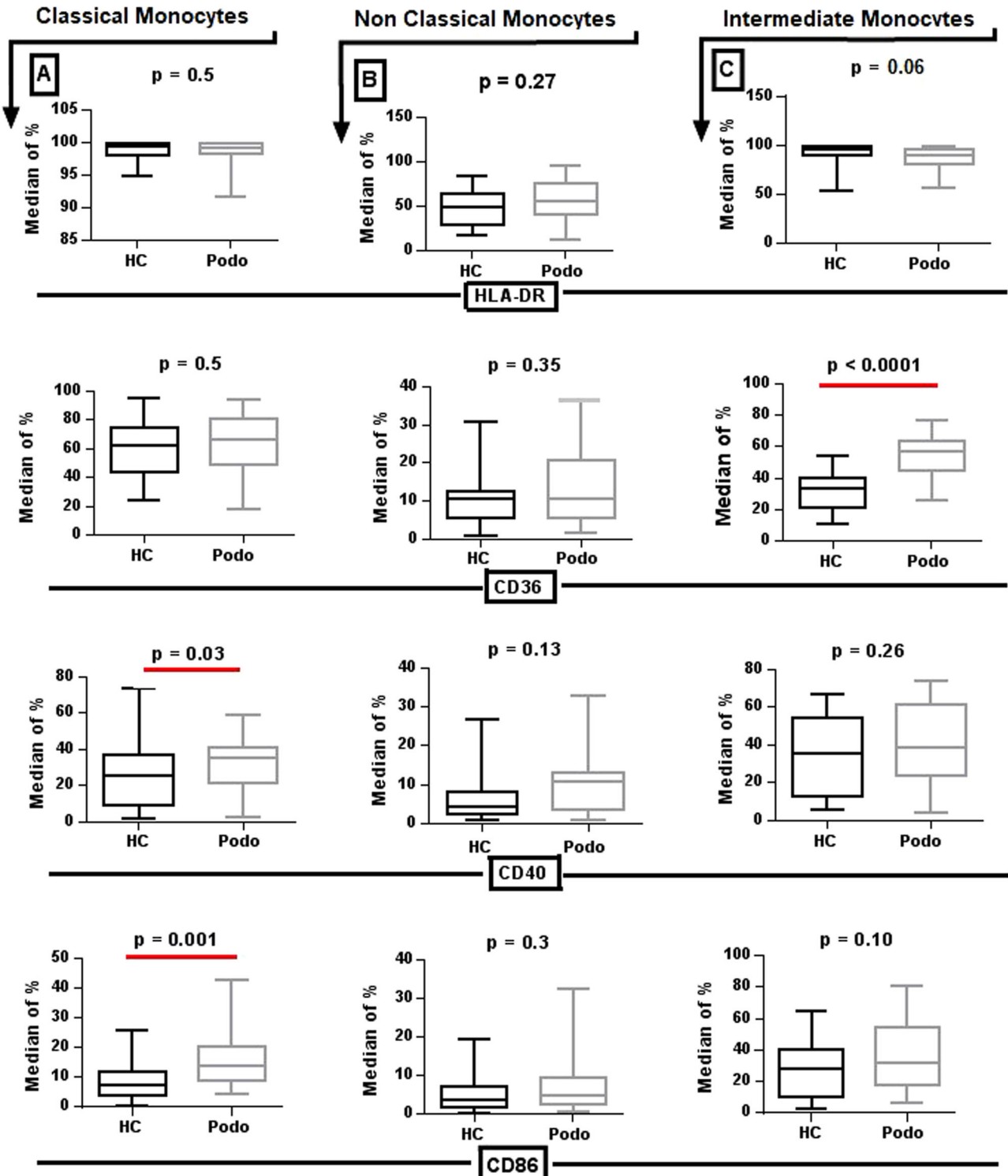

**Fig. 2 | Median expression of HLA-DR, CD36, CD40, and CD86 markers on monocyte subsets from podoconiosis patients and healthy controls.** The figure shows box plots depicting median, interquartile range, minimum, and maximum values of subset frequencies in monocytes defined from peripheral blood cells of 43 podoconiosis patients and 34 healthy controls. Columns (**A**–**C**) represent classical, non-classical and intermediate monocyte subsets respectively. Each row represents one of the four markers: the top row shows results for HLA-DR, the second shows results for CD36, the third row shows results for CD40 and the bottom row shows results for CD86. *P* values were derived using the Mann–Whitney *U* test of two-sided independent *t* test. Source data that is used to generate this graph is provided as a "Source Data" file Figs. 1–3.

podoconiosis patients while histones and cell division transcripts were significantly downregulated.

Variation in HLA haplotypes or alleles has long been associated with susceptibility or resistance to infectious and non-infectious diseases including autoimmune diseases and drug and contact hypersensitivity reactions. Haplotypes or alleles with potent antigen processing and presentation potential are selected to protect against pathogens and passed on to the next generation[18]. However, such

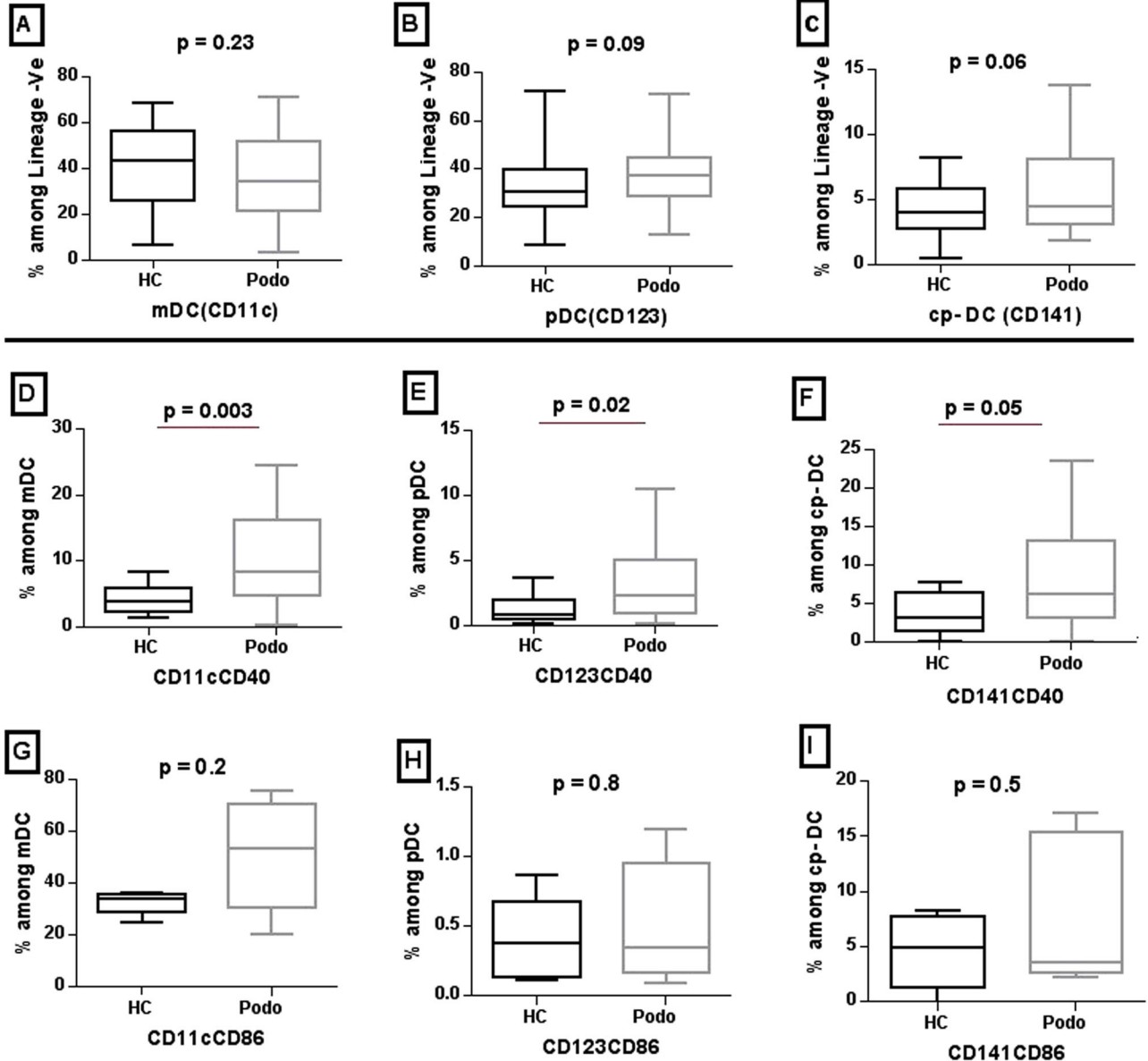

**Fig. 3 | Proportion of dendritic cell subsets and median expression of CD40 and CD86 markers on the three DC subsets from podoconiosis cases and healthy controls.** Dendritic cell (DC) subsets were sub-enumerated from lineage-negative and HLA-DR positive gated populations; (**A** myeloid DC, **B** plasmacytoid DC, and **C** cross-presenting DC). **D**–**F**, **G**–**I** shows expression of CD40 and CD86 markers among these DC subsets, respectively. The box plots represent median, interquartile range, minimum and maximum values for subset frequencies and the activation markers defined from peripheral blood cells of 43 podoconiosis (Podo) patients and 34 healthy controls (HC). *P* values were derived using the Mann–Whitney *U* test of two-sided independent *t* test. Source data that is used to generate this graph is provided as a "Source Data" file Figs. 1–3.

highly evolved haplotypes associated with efficient immune responses can also target self-antigens and cause autoimmunity or react to foreign antigens that are not pathogen-derived such as metals (e.g., contact with nickel) or drugs such as abacavir[19]. While some HLA-associated autoimmune diseases are characterized at the level of self-peptide presentation by the implicated gene products[20], mechanisms underlying HLA associations in many autoimmune diseases are not clearly understood. This is in part due to extensive linkage disequilibrium across the region that can span more than 2 Mb[21] and complicates elucidation of the functional contributions of specific variants. For other HLA-associated autoimmune diseases, it is clear that disease pathology is induced by presentation of altered self-antigens or foreign antigens that cross react with self-antigens due to molecular mimicry[22,23].

However, not all autoimmune or hypersensitivity reactions involving exogenous factors involve mimicry—examples include coeliac disease[24] and chronic beryllium disease. Chronic beryllium disease (CBD) is a lung disorder caused by chronic exposure to beryllium and the development of a specific immune response in genetically susceptible individuals. The disease is associated with an HLA-DP variant with a beta chain polymorphism defined by the presence of negatively charged glutamic acid at position 69. This allows beryllium to bind to the HLA-DP molecule to initiate a beryllium-specific polyclonal T-cell response leading to inflammation and tissue damage[25,26]. A prototypical HLA-mediated hypersensitivity reaction is a response to the drug abacavir. Abacavir hypersensitivity is specifically induced by the binding of abacavir to an HLA-B*57:01 allele. This binding cause a change in chemical and structure of the antigen binding pocket of the

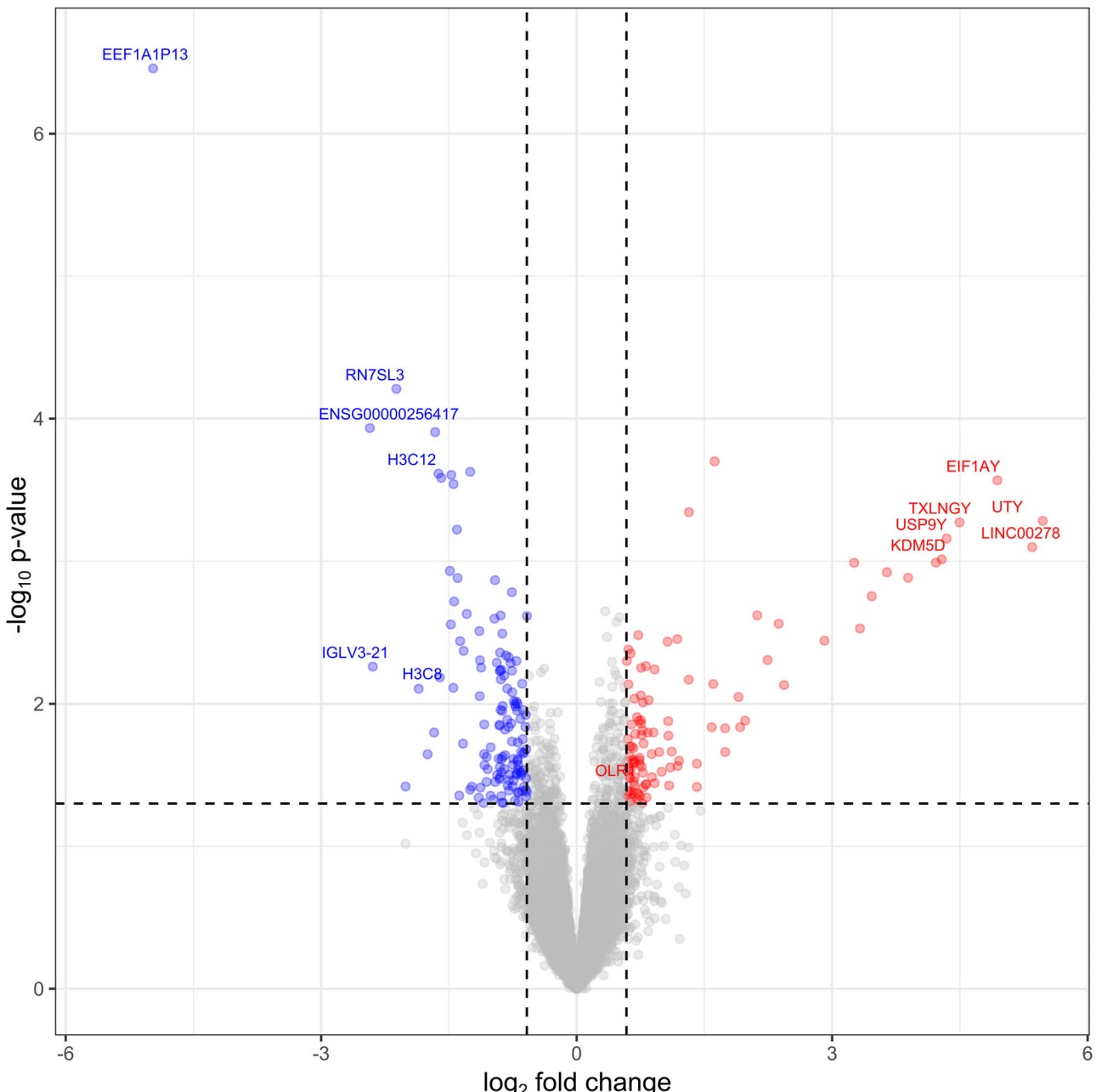

**Fig. 4 | Volcano plot showing fold change and *P* value for differentially expressed genes in podoconiosis compared to healthy controls.** Volcano plot analysis of differentially expressed genes between 19 podoconiosis patient and 15 healthy control samples. The log2-fold change values are plotted on the *x* axis and compared with the negative log10 *P* values on the *y* axis. Blue dots represent significantly downregulated genes, and red dots represent significantly upregulated genes in podoconiosis patients compared with healthy controls with a fold change of >1.5 and *P* < 0.05. The *F* test was used for deriving *P* values and adjustments were made for multiple comparison. Source data that is used to generate this graph is provided as "Source Data" file Figs. 4 and 5.

HLA molecule, thereby altering the repertoire of self-antigens that binds to the HLA pocket. This eventually drive the binding of new self-peptides that activate polyclonal CD8 T-cell response[27]. Given the role of soil exposure, it is possible that an exogenous element or mineral particles incorporating an immunogenic element could similarly interact with a particular HLA molecule in podoconiosis. This could be further investigated through analysis of the T-cell receptor repertoires in podoconiosis patients.

T-cell expression of HLA-DR along with other activation markers such as CD38, CD69 and Ki-67, is known to be induced after activation by antigens or mitogens[28]. For example, increased expression of activation markers is commonly seen in acute infections, and other chronic inflammatory conditions of either infectious origin such as

tuberculosis[29] and HIV[30], or autoimmune origin such as rheumatoid arthritis[31] and systemic lupus erythematosus[32]. However, there is little direct evidence that such activated cells are antigen-specific. In fact, in studies of HIV in particular it would appear that the vast majority of activated T cells are not HIV-specific[33], raising the question as to whether expression of activation markers in chronic diseases necessarily reflects a presumed antigen induction.

It is widely accepted that HLA-DR expression represents a state of cellular activation. But there are some discrepancies on its expression and functional role on human T cells in the literature. One study showed increased expression of HLA-DR on T cells enhanced memory pool generation and activation of cytotoxic CD8 T cells for anti-tumor responses through T-cell–T-cell synapse formation and IFN-γ

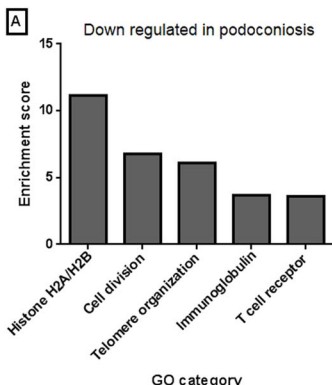

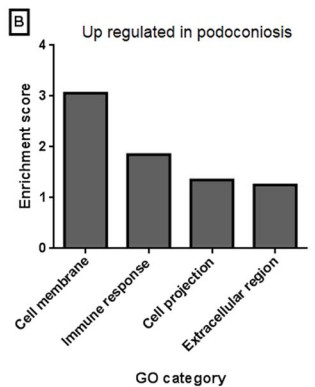

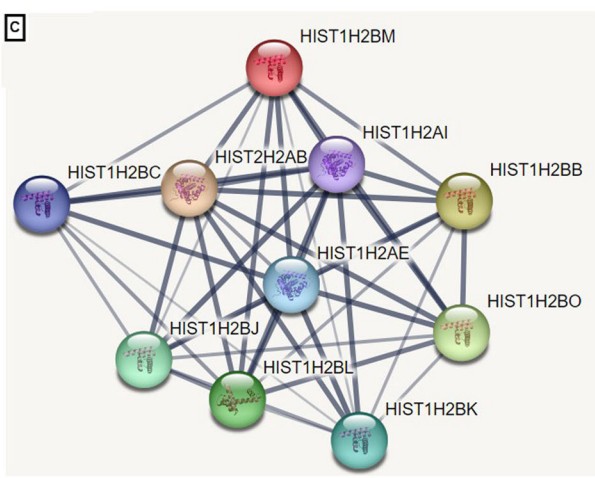

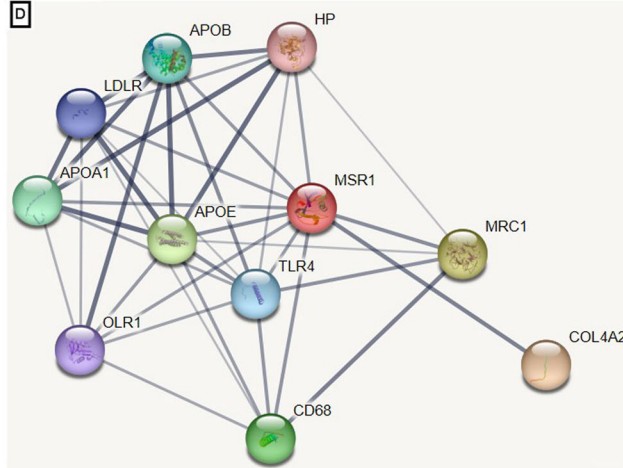

**Fig. 5 | Functional enrichment and protein–protein network analysis for the top upregulated and downregulated genes in podoconiosis patients relative to healthy controls.** Most enriched categories for downregulated (**A**) and upregulated (**B**) genes in podoconiosis patients. The *y* axis represents enrichment scores as −log10 (*P* value) and the *x* axis indicates the gene ontology (GO) category enriched in the pathway analysis. Gene names from the most enriched GO category were submitted to the STRING database for the downregulated (**C**) and upregulated genes (**D**) separately and a protein network interaction was generated for the given enriched pathway. Here nodes represent proteins with different colors representing the different submitted gene lists and additional proteins with interaction with the submitted gene lists. The lines represent known and predicted interactions between proteins, with the thickness of the line indicating the strength of interaction. The *F* test was used for deriving *P* values and adjustments were made for multiple comparison. Source data that is used to generate this graph is provided as "Source Data" file Figs. 4 and 5.

secretion[34]. The higher HLA-DR expression on fresh PBMC T cells in podoconiosis patients could be due to bystander activation from the pro-inflammatory cytokine milieu. Consistent with this possibility, we observed higher levels of TNF-α and IL-1β in unstimulated culture wells and IL-1β mRNA in the peripheral blood of podoconiosis patients (manuscript in preparation).

The expression of CD62L in podoconiosis patients was significantly lower compared to healthy controls further highlighting the activated state of T cells in podoconiosis patients. Reduced CD62L expression is a feature of effector memory T cells and following CD62L shedding, memory T cells re-enter the circulation from lymph nodes where they can exert their effector function[35]. The recruitment of T cells to inflamed sites and their effector function is greatly enhanced after shedding of CD62L and acquisition of selectins such as CD62E and CD62P to attach to the endothelial cells wall[36]. Immunohistochemistry analysis of Price's archival lymph node samples showed that CD4 T lymphocytes were the predominantly infiltrating cells from podoconiosis patient biopsy samples, suggesting these T cells could play a role in development of the disease[37].

There was no significant difference in the distribution of the three monocytes subsets (classical, intermediate and non-classical) between podoconiosis patients and healthy controls. However, the expression of the co-stimulatory molecules CD86 and CD40 among classical monocytes was higher in podoconiosis patients. The three DC subsets had significantly higher levels of CD40 expression while expression of CD86 was higher only in myeloid DCs in podoconiosis patients compared to the healthy controls. The classical monocytes and myeloid DCs are the main subsets in each lineage which are involved in production of inflammatory cytokines and reactive oxygen species during infection or recognition of ligands by their pattern recognition receptors[38,39]. In other chronic inflammatory HLA-associated diseases such as multiple sclerosis[40] and coeliac disease[41], monocytes have been shown to have increased expression of CD40, CD86 and HLA-DR in ex vivo stimulation assays compared to healthy controls. Similarly, immunohistochemistry analysis in Crohn's disease and ulcerative colitis patients showed the levels of activation markers like CD83 and CD86 expressed on DCs, primarily mDCs, were significantly higher in inflamed mucosa from patients compared with non-inflamed mucosa and healthy controls[42,43]. It has been suggested that mature DCs with higher levels of activation marker expression are likely to play a key role in mediating inflammation at the site of pathology[43]. Dendritic cells are strategically and abundantly located under the skin sub-mucosa and they could be the first immune cells to be exposed to potential foreign antigens or particles that get access through the skin in podoconiosis[44]. In the current study we have only studied cells from peripheral blood, targeted analysis of the local microenvironment

**Table 1 | List of significantly upregulated genes from the top enriched clusters based on their gene ontology category, in peripheral blood from 19 podoconiosis patients and 15 healthy controls**

| GO category | Upregulated gene | Gene description | Fc Podo vs HC | P value |
|---|---|---|---|---|
| Immune response | CD1A | CD1a molecule | 1.7 | 0.03 |
| | CD80 | CD80 molecule | 1.2 | 0.07 |
| | CD86 | CD86 molecule | 1.3 | 0.06 |
| | HLA-DQB1 | HLA-DQ beta molecule | 1.8 | 0.3 |
| | WAKMAR | Wound and keratinocyte-associated lnRNA | 1.57 | 0.028 |
| Cell membrane and signaling | OLR1 | Oxidized low-density lipoprotein | 1.74 | 0.03 |
| | CD163L1 | CD163-like molecule 1 | 1.57 | 0.02 |
| | MSR1 | Macrophage scavenger receptor 1 | 1.6 | 0.05 |
| | ADIPOR1 | Adiponectin receptor | 1.54 | 0.04 |
| Cell projection | ITGA1 | Integrin subunit alpha 1 | 1.3 | 0.04 |
| | FFAR4 | Free fatty acid receptor 4 | 1.6 | 0.03 |
| Extracellular region | CTSB | Cathepsin B | 1.7 | 0.01 |
| | MPO | Myeloperoxidase | 1.7 | 0.04 |
| | EREG | Epiregulin | 2.5 | 0.006 |
| | BNIP3L | BCL2 interacting protein 3 like | 1.57 | 0.04 |

*GO* gene ontology, *Fc* fold change, *HC* healthy controls, *vs* versus.
Gene names are italicized.

**Table 2 | List of significantly downregulated genes from the top enriched clusters based on their gene ontology category, in peripheral blood from 19 podoconiosis patients and 15 healthy controls**

| GO category | Downregulated gene | Gene description | Fc Podo vs HC | P value |
|---|---|---|---|---|
| Histones | H3C12 | H3 clustered histone 12 | 3.1 | 0.0002 |
| | H2AC12 | H2A clustered histone 12 | 3 | 0.0002 |
| | H2BC14 | H2B clustered histone 14 | 2.8 | 0.001 |
| | H3C8 | H3 clustered histone 8 | 3.1 | 0.0002 |
| Cell division cycle | CDC6 | Cell division cycle 6 | 2.3 | 0.0001 |
| | CDCA5 | Cell division associated 5 | 2.6 | 0.003 |
| Immunoglobulin domain | IGLV3–21 | Immunoglobulin lambda variable 3–21 | 4.9 | 0.005 |
| | IGLV1–47 | Immunoglobulin lambda variable 1–47 | 3.5 | 0.007 |
| | IGLC3 | Immunoglobulin lambda constant 3 | 3 | 0.006 |
| | IGLC2 | Immunoglobulin lambda constant 2 | 2 | 0.03 |
| T-cell receptor | TRAV26-1 | T-cell receptor alpha variable 26-1 | 1.57 | 0.03 |
| | TRAV10 | T-cell receptor alpha variable 10 | 1.74 | 0.04 |
| | TRBC1 | T-cell receptor beta constant 1 | 1.75 | 0.03 |

*GO* gene ontology, *Fc* fold change, *HC* healthy controls, *vs* versus.
Gene names are italicized.

could reveal more insight into which cells and subsets are mainly localized in the inflamed areas.

The RNA-sequencing result showed 108 genes were significantly upregulated in podoconiosis and functional enrichment analysis for these genes showed the cell membrane and immune response clusters had the highest enrichment score in the pathway analysis. The upregulated immune response genes like *CD80, CD86, CD1A*, and *HLA-DQB1* correlated with the PBMC surface immunophenotyping data where classical and myeloid DC also exhibited higher expression of these markers. Further review of the 'immune response' GO cluster and genes within this category (*CD1A, CD80, CD86*, and *HLA-DQB1)* revealed related pathways which were in the same biological process associated with numerous autoimmune and infectious diseases. Many of these diseases themselves are caused by complex interactions between environmental and host genetic factors, including HLA gene variants.

The upregulation of *HLA-DQB1* in this study is of interest due to the documented association between this gene (as well as *HLA-DRB1*) and podoconiosis[9,10]. Variation in the *HLA-DQB1* gene and its upregulation have been linked with susceptibility to different solid organ and tissue fibrosis[45,46]. A recent study in 2022 by Zhou et al. which integrated RNA-Seq with GWAS data indicated a SNP (rs9273410) in the *HLA-DQB1* region was significantly associated with increased susceptibility to silicosis[47]. Silica particles have been identified in macrophages and lymph nodes in people living in podoconiosis-endemic regions[11] although evidence directly implicating silica in its pathogenesis is currently lacking. However, none of these studies, including the current study, elucidated the potential epitope or the mechanism by which *HLA-DQB1* contribute towards silicosis or the development of fibrosis. Hence, further studies to elucidate the function and regulation of this region in such pathologies are warranted.

Variation in HLA genes has been associated with a number of infectious diseases, raising the possibility that infection may play a role in podoconiosis. However, the available evidence is more consistent with a noncommunicable etiology and there is no evidence of direct involvement of a pathogenic organism. Recent studies which analyzed the skin microbiome of podoconiosis patients reported the presence of distinct bacterial species[48,49]. Superficial skin infections leading to acute dermatolymphangioadenitis attacks could exacerbate the lymphoedema and cause local inflammation but this tends to occur later in the disease, once lymphoedema is established[49]. Therefore, it is conceivable that skin microbiome changes may also contribute to disease progression and the increase in activation markers in podoconiosis patients, but are unlikely to be involved in initial pathogenesis. None of the study subjects had acute dermatolymphangioadenitis at the time of enrollment.

Genes which were significantly upregulated in the cell membrane and signaling GO clusters were mainly from the scavenger receptor family including *OLR1*, *CD163L1*, and *MSR1*. These receptors are expressed on different cells such as endothelial cells, monocytes and macrophages. In line with this higher level of CD36 was observed in intermediate monocytes subsets of podoconiosis patients from the PBMC immunophenotyping in the current study. Similar to the CD36 scavenger receptor, these receptors have a broad range of ligands including oxidized lipoproteins, heat shock proteins, asbestos, and silica[50,51].

Phagocytosis of silica by macrophages leads to impaired lysosomal degradation because of the particulate nature of the silica. This pathway of silica absorption in macrophages derived from murine cell line was linked with induction of inflammatory cytokines, reactive oxygen specious, apoptosis and fibrosis[50,52]. It was suggested CD36 contributes to sterile inflammation via internalization of components like oxidized lipoproteins and cholesterol crystals by assembling the NLRP3 inflammasome complex and secretion of pro-inflammatory cytokines[53]. Podoconiosis patients may have a higher rate of silica binding and retention compared to healthy controls which could be mediated by higher expression of these scavenger receptors. Of course, once internalized, differences in responses to the minerals between patients and healthy controls could also play a role in susceptibility to developing the disease (it has already been shown in the current study that there were substantial immunological, lysosomal enzyme and protein differences between the two groups). Price's elemental analysis indicated some differences in the ratio of aluminum to silica and birefringent particles (uncoated particles) found in lymph node samples from patients relative to healthy controls[11].

Higher levels of transcripts from the genes encoding the BCL interacting protein 3 like protein (*BNIP3L*) which is involved in activation and assembly of the apoptosome protein complex domains[54], cathepsin B (*CTSB*) which is involved in lysosomal protein degradation, processing and presentation as well as extracellular matrix degradation[55], and myeloperoxidase (*MPO*) which is involved in reactive oxygen species production[56,57] were observed in podoconiosis patients compared to healthy controls. Similarly, higher levels of transcripts from the genes encoding epiregulin (*EREG*), wound and keratinocytes migration associated lnRNA (*WAKMAR*) which are involved in tissue and wound healing and integrin subunit alpha 1 (*ITGA1*) which is a cellular adhesion molecule for collagen and laminin[58,59] were observed in podoconiosis patients compared to healthy controls. The binding of collagen and laminin in the extracellular matrix through ITGA1 could contribute in the pathology of podoconiosis through enhancing the fibrosis. Although the role of silica in the immunopathogenesis of podoconiosis is still unclear, it remains possible that silica (and/or other soil mineral) induces inflammation leading to the release of reactive oxygen species and apoptosis of cells[52,53,58] which could lead to pathology through disruption of lysosomes and the upregulation of genes like *MPO*, *CTSB*, *ITGA1*, and *BNIP3L* which directly or indirectly shape the inflammation and the extracellular matrix. However, further research to identify the soil trigger in podoconiosis is still required.

The significantly downregulated genes in podoconiosis patients were associated with the GO categories such as; nucleosome, histone, cell division, immune globulin domain and TCR receptor domains. Of these enrichment clusters the histones had the highest enrichment score in the pathway analysis followed by cell division clusters. Histones are the major structural component of the nucleosome. Modification of histones through deacetylation or methylation is a fundamental aspect of gene regulation by altering access to the binding of transcription factors or polymerases. Histone modification plays a major role in overall cell division and transcription of genes[60]. A review by Kerstin Klein and Steffen Gay indicated DNA methylation and posttranslational histone modifications in particular in synovial fibroblasts play a role in the development of rheumatoid arthritis[61]. It is not clear how this modification might play a role in podoconiosis, hence future studies are needed to elucidate the role of epigenetic modification in podoconiosis.

The downregulation, in particular in the TCR receptor genes was expected given that PBMC immune-phenotyping showed a relatively higher level of expression of activation markers in podoconiosis patients. TCR downregulation following persistent TCR ligation is one mechanism of limiting hyperactive immune responses after pathogen control[62]. It is not clear if this downregulation was clone-specific or due to a bystander immune exhaustion effect due to the chronic nature of the disease. Targeted analysis of the TCR repertoire to characterize T-cell clonality or TCR usage would be interesting to investigate in more detail in podoconiosis.

In conclusion the high level of activation markers on T cells, classical monocytes and mDCs suggest that these subsets could play a central role in priming and driving the immune response in podoconiosis patients, keeping in mind that the latter two subsets are the main producers of inflammatory cytokines. Moreover, upregulated levels of immune activation, inflammatory enzymes and scavenger receptor transcripts suggest persistent inflammation and impaired adipose tissue metabolism could be taking place in the lower legs of podoconiosis patients. This could progressively lead to development of a fibrotic microenvironment and impaired lymphatic function. In the current study we have made progress in describing and understanding the immune response in podoconiosis, yet specific driving pathways and the causative agent(s) remain to be elucidated. Replication and validation of these findings observed in podoconiosis patients by corroborating with data from tissue biopsy analysis could ultimately lead to potential treatment options within these pathways, which could ameliorate the disease symptoms and progression.

## Methods

### Ethical approval and informed consent

A rapid ethical appraisal was conducted using in-depth interviews and focus group discussion prior to sample collection to tailor the informed consent process to the local socioeconomic norms and to address the participants' concerns before undertaking the research[63]. Ethics approval was obtained from the AHRI/ALERT (Protocol No. PO-3818) and Ethiopian National Science and Technology (Ref N0. AH/00229/001241/21) ethics review committees in Ethiopia and the BSMS Research Governance and Ethics Committee in the UK (Ref. No. ER/BSMS9DJB/1). The study participants gave written consent before sample collection. Clinical and biological data obtained from study participants were kept secure and confidential as per the data sharing and protection agreement between AHRI and BSMS partners.

### Study population and design

A community-based case-control study design was employed to enroll podoconiosis patients and healthy control individuals from northeast

Ethiopia. The study was conducted in Bahir Dar and neighboring districts which is a podoconiosis-endemic area. Podoconiosis cases were selected from two health centers in the Bahir Dar district. The cases were already diagnosed and staged by trained nurses and health officers using a validated clinical algorithm[64] and the Tekola staging system[17] and were registered in the health center log books. We deliberately selected podoconiosis cases which were stage two and three. Stage 1 podoconiosis patients were excluded due to diagnostic difficulties as the clinical features of early podoconiosis could be confused with a range of other conditions such as heart or liver failure that also lead to ankle swelling. Late stages (stages 4 and 5) were also excluded as in these advanced stages of disease fibrosis predominates and any active inflammation involved in the pathogenesis of podoconiosis may have "burnt out". Unrelated sex and age-matched healthy controls who were over the age of 18 years old, had lived in the same study area as the patients at least for 10 years, who had no family history of podoconiosis and had never worn shoes consistently, and were therefore exposed to the soil without developing podoconiosis, were selected. Individuals who had underlying chronic disease like diabetes, liver, kidney or cardiac disease, or who were taking medication were excluded from the study. Moreover, individuals who were unwell on the day of sample collection (for example, due to intercurrent infection), cases who had clinical evidence of secondary skin infection or acute dermatolymphangioadenitis attack (ADLA), those with known HIV infection or who tested positive for HIV with screening tests (based on the national testing algorithm) were also excluded.See Supplementary Table 1 for sociodemographic summary).

## Specimen collection and laboratory assays
Whole blood was collected from a total of 64 podoconiosis patients and 49 healthy controls using heparinised tubes for immunological assays and using PAXgene tube for transcriptimics studies. Peripheral blood mononuclear cells (PBMCs) were isolated by Ficoll-Hypaque (Sigma-Aldrich, Catalog no. GE17-5442-03) density gradient centrifugation from the heparinised peripheral blood. The PBMCs were used for immunophenotyping studies of T cells, monocytes and dendritic cells by using multicolor flow cytometry. For the transcriptomics assay 24 samples were selected from each study group by matching them based on their age, sex, PBMC number and RNA quantity and quality. PBMC were isolated, stained and paraformaldehyde fixed at the study site laboratory located on average 15 km from the sample collection site. The fixed samples were stored at -80°C with freezing media until transportation. Fixed samples were air-transported within two weeks of processing for flow cytometry acquisition at AHRI in Addis Ababa. The libraries were also prepared in AHRI and shipped to UK, Leeds University for sequencing.

## Flow cytometry
The freshly isolated PBMCs were stained with monoclonal antibodies targeting the respective markers for each cell lineage. All antibodies were from Beckton Dickinson (BD) unless specified. T cells (PBMC of ~$2 \times 10^5$) were stained with CD3-APC-H7, CD4-BV510, CD8-BV421, CD38-APC, HLA-DR-PE-Cy7, Ki-67-PerCPCy5.5, CD62L-PE, monocytes (PBMC of ~$4 \times 10^5$) were stained with CD16-FITC, CD14-BV421, CD40-PE, CD86-BV510, CD36-PerCP-Cy5.5, HLA-DR-APC-Cy7, dendritic cells (PBMC of ~$4 \times 10^5$) were stained with CD11c-BV421, CD123-PerCP-Cy5.5, CD141-APC, CD40-PE, CD86-BV510, HLA-DR-APC-Cy7, and a lineage cocktail comprised of FITC conjugated antibodies to CD3, CD14, CD16, CD19, CD20 and CD56. Intracellular staining for Ki-67 was performed by permeabilising cells with Cytofix/Cytoperm™ (BD, Catalog no. 554714). All data were acquired using a FACS Canto II flow cytometer (BD Biosciences, San Jose, CA, USA). Unstained and compensation controls were run during every acquisition. A minimum of 100 000 events for T cells and 200 000 events for monocytes and dendritic cells were collected for each analysis. All data were analyzed using the FlowJo software (Mac Version 9.6, TreeStar Inc, USA).

## Statistical analysis
The non-parametric Mann–Whitney $U$ test was applied to assess the difference in the expression of different surface and intracellular biomarkers and the median MFI because in general distributions were not observed to be normal for most of the activation markers. However, we also compared marker expressoin using mean MFI values and got very similar $P$ values between podoconiosis patients and healthy controls. GraphPad Prism V-6 (GraphPad Software Inc., CA, USA) was used to calculate significance levels using two-sided tests where $P$ value of less than 0.05 was considered statistically significant.

## RNA extraction, library preparation and sequencing
RNA was extracted from samples collected in PAXgene tubes using MagMAX™ bead-based extraction method (Life Technologies, Catalog no. 4451894). The extracted RNA was checked for quantity and quality using the Qubit™ RNA high-sensitivity assay kit (Invitrogen, Catalog no. Q32855) and the dsDNA high-sensitivity assay kit (Invitrogen, Catalog no. Q32854). A total 0.5 µg RNA samples from 24 podoconiosis and 24 healthy control samples were then used for preparing the sequencing libraries. The libraries were prepared using strand-specific fast select library preparation kit based on the manufacturer's instruction (Qiagen, Catalog no. 180450). Briefly, RNA was fragmented and ribosomal and globin RNAs removed by incubating it in decreasing temperature gradient using QIAseq FastSelect rRNA and globin mRNA removal kit" (Qiagen, Catalog no. 335376). The whole sample from the fragmentation step was used for first-strand synthesis using reverse transcriptase buffer mix. Second strand synthesis was carried out using 2$^{nd}$ strand buffer and enzyme mix. The 5′ phosphorylation during 2nd strand synthesis allows subsequent strand-specific ligation with the Illumina Dual-Index Y-Adapters. The 24-plex single-use adapter plate was used for the current library preparation. Following adapter ligation the libraries were subjected to PCR amplification cycles. Finally, the libraries were cleaned based on their size using blue pippin (Sage Science) and sequenced in two runs using the Illumina Next-Seq500 NGS platform.

## Quality controls and trimming the sequencing reads
The quality of the sequencing was assessed based on the Phred scale using FatQC V 0.11.9, and almost all reads have base qualities above Phred score of 30, which represent an incorrect call rate of 1 in 1000 bases, giving a base call accuracy of 99.9%. The quality of the reads, prior to and post-trimming with cutadapt for one representative sample is shown in Supplementary Fig. 5A, B, and the overall quality score for all samples generated by MultiQC V1.13. before and after trimming is presented in Supplementary Fig. 6C, D.

## Mapping reads to the human reference genome, sorting and indexing the mapped reads
Once the reads were controlled for quality and the short inserts trimmed, a splice-aware program HISAT2 v2.2.0 was used to align the reads to the reference human genome (Ensembl GRCh38). A program named SAMtools v0.1.20 was used to compress the aligned reads to a binary format (.bam), index and sort them based on chromosome order. The sorted.bam file was used by the program featureCounts to count the number of reads that mapped to the reference human genome. The alignment rate was very good, whereby ~95% of the reads for all samples mapped to the reference genome in the current study (Supplementary Tables 3 and 4 presents summarized output from feature counts including the total number of reads mapped, primary mapped reads, assigned reads, adapter reads and the overall alignment rate for healthy control and podoconiosis samples, respectively).

### Differential gene expression (DGE)

The counted reads from featureCounts were imported to R (V4.4.3) and DGE was performed using edgeR packages. Prior to performing the DGE analysis, the reads were filtered to remove genes below the set parameter and normalized by assigning a normalization factor for each sample. Reads with more than 5 million counts were included in the final DGE analysis. The potential effect of age and sex differences in the DGE was also controlled by fitting these variables into the analysis model in edgeR. A $P$ value of less 0.05 from the negative log10p was considered significant.

### Biological functions and pathway enrichment analyses

To further explore the biological functions and pathways to which the differentially expressed genes belong, the list of differentially expressed genes was submitted to an online gene ontology (GO) analysis site. The GO and functional enrichment analysis was performed using Database for Annotation, Visualization and Integrated Discovery (DAVID; https://david.ncifcrf.gov/). The list of differentially expressed genes were submitted online to DAVID with their Ensemble gene ID. Pathway enrichment analysis was performed by selecting default terms from the following databases: Kyoto Encyclopedia of Genes and Genomes (KEGG), Gene ontology terms in biological processes (BP), cellular component (CC), molecular function (MF) Canonical Pathways (CP), and the Comprehensive Resource of Mammalian protein complexes (CORUM).

### Construction of protein–protein interaction network

To assess the downstream interaction of the differentially expressed genes in biological processes, protein–protein interaction (PPI) network analysis was performed using the Search Tool for the Retrieval of Interacting Genes/Proteins (STRING) database (http://string-db.org/). A network was built from our data by submitting gene lists from the highly enriched clusters and the lowes $P$ value for the upregulated and downregulated genes separately. Protein interaction networks were built for the different enriched clusters based on significantly differentially expressed genes from Tables 1 and 2 (see "Results").

### Reporting summary

Further information on research design is available in the Nature Portfolio Reporting Summary linked to this article.

## Data availability

All available data are included in the manuscript and the supplementary file. Data sources used to generate graphs is provided with this paper as 'Source Data'. The processed RNA-Seq data is also provided as a source data along with the raw RNA-sequencing data which has been deposited in ArrayExpress (https://www.ebi.ac.uk/arrayexpress/) under the accession number E-MTAB-13860. Source data are provided with this paper.

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

## Acknowledgements

The authors thank the Amhara regional health bureaus and the Amhara Public Health Institute for their support with participant enrollment, data and sample collection in Ethiopia. The authors like to thank Dawit Hailu at AHRI for his assistance during library preparation. The authors also thank the study participants who volunteered to participate in this study. This work was supported by the National Institute for Health Research (NIHR) Global Health Research Unit on NTDs at Brighton and Sussex Medical School using Official Development Assistance (ODA) funding [Grant number GHR 16/136/29], grant received by M.J.N. and G.D. The views expressed here are those of the author(s) and not necessarily those of the NIHR or the Department of Health and Social Care.

## Author contributions

M.N.: sample collection, data analysis, figure production, and writing the first draft of the manuscript. M.C.: study conception and fieldwork supervision. T.G.: sample collection and data analysis. F.A.: sample collection and data analysis. D.Alc.: data analysis, writing manuscripts, and interpretation. B.T.: data analysis and interpretation. D.Alt.: data analysis and interpretation. R.B.: data analysis and interpretation. G.D.: study design and manuscript write-up. R.H: study design, data interpretation, manuscript writing, and Co-PI in Ethiopia. M.J.N.: study conception and design, writing the manuscript, PI for funding, and overall study. All authors edited the manuscript and approved the final version.

## Competing interests

The authors declare no competing interests.
