## [Peer Review File · Nature Communications]

Evidence for immune activation in pathogenesis of the HLA class II associated disease, podoconiosisREVIEWER COMMENTS

Reviewer #1 (Remarks to the Author):

Negash et al. characterised activation markers of monocyte and T and dendritic cell subsets from individuals that suffer from lymphedema caused by contact to irritant volcanic soil (podoconiosis). This study was based on genome-wide studies showing that disease susceptibility is associated with variation in HLA class II genes. Consequently, the authors immunophenotyped immune cells (e.g., HLA-DR, CD38, CD62L and CD40, CD86, CD36, respectively) as well as performed RNA sequencing gene expression analysis and revealed that receptors and activation markers involved in antigen processing and presentation and inflammation are increased in podoconiosis patients in comparison to healthy controls. I congratulate the authors for the very interesting results which are of great interest for researchers dealing with this disease, especially the pathogenesis of this stigmatizing disease remains unknown. However, some issues need to be resolved before I can endorse publication.

1) In total the authors analysed 64 podoconiosis and 49 healthy individuals, but analysed only 56 vs 44 (T cells), 43 vs 34 (monocytes/dendritic cells) and 21 vs 23 (transcriptomic). Can the authors explain the selection (maybe present a flow chart). Are the results consistent when only the 21 vs 23 samples analysed in the flow cytometry approach?

2) In regards to 1), in my opinion it would be necessary to show an overview table about the study characteristics including gender, age and lymphedema stage (do the authors have information about ADLA?). Were all results controlled for a potential bias of age and gender?

3) In regards to the gating strategies, did the authors include a live and dead staining, which is very important when activation markers are analysed.

4) Figure 1: Were there any differences in the CD3 and/or CD4 cell number/frequencies per se? The authors mentioned that also MFI show the same pattern as the frequencies; is this also true for the cell numbers (this would prove that the findings are consistent and solid).

5) Figure 2/Supplementary Figure 2: Since the authors state in the main figure classical, non-classical and intermediate monocytes these phrases should be linked to the gating strategy. The gating strategy for the intermediate monocytes (CD16+CD14+) is missing.

6) Can the authors explain the choice of the different activation markers? Especially, why CD80 was not analysed, which has been found upregulated in the transcriptomics analysis and is important for T cell activation.

7) The authors state that HLA expression/activation is associated with immunity against pathogens and nicely described the associations with silica crystal uptake in regards to the pathogenesis of the disease, but as mentioned above it has been shown (e.g., in lymphatic filariasis lymphedema patients) that bacteria can drive acute dermatolymphangioadenitis attacks (ADLA) promoting the progression of the lymphedema. Recently, two publications analysed the skin microbiome of podoconiosis patients and associated distinct bacteria spp. and change of normal skin flora to high lymphedema stage legs. Thus, involvement of bacteria/pathogens in the progression of the disease should be discussed as well, especially it nicely fits to the obtained activation patterns of the different immune cells. Maybe it is a combination of silica uptake which induces inflammation and destruction of the skin barrier allowing the entrance of pathogens driving ADLA and consequently disease progression.

8) Line 560: should be supplementary table 2 and 3

Reviewer #2 (Remarks to the Author):

Will the work be of significance to the field and related fields?

Yes

How does it compare to the established literature? If the work is not original, please provide relevant references.

It adds new insights through the RNA transcription data and confirms previous data.

Does the work support the conclusions and claims, or is additional evidence needed?

Yes

Are there any flaws in the data analysis, interpretation and conclusions?

Some comments are added in the uploaded file to verify the right statistical analysis were conducted.

Do these prohibit publication or require revision?

Minor revision

Is the methodology sound?

Yes

Does the work meet the expected standards in your field?

Yes

Is there enough detail provided in the methods for the work to be reproduced?

Suggestions were made in the uploaded material

Pasting my comments (minus the two figures here as well)

Negash et al. present single cell-based data for individuals with podoconiosis, which is an understudied preventable disease associated with exposure to minerals found in volcanic soils. Silica particles penetrating the skin trigger an inflammatory response and the characteristic swelling in the foot. The authors use healthy endemic individuals as their control group, which is a plus to reduce other potential environmental factors contributed by people living elsewhere in the world. If they had used instead a European or American control group significant differences might have been identified that were unrelated to podoconiosis. However, one could use publicly available transcriptomics data e.g. from the genome-based tissue expression consortium (PMID 25809799) to cross-check the control group. Elevated transcripts in the control group that also show differences in the podoconiosis population are likely unrelated to the disease. A quick check would be to verify that their biomarkers identified to be associated with podoconiosis hold true when testing the healthy population against a previously published reference transcriptome of other healthy individuals not living in the endemic area.

Using peripheral blood the authors identify several markers allowing them to categorize and distinguish healthy from early and late stage podoconiosis patients. Previous genome-wide studies in Ethiopia showed association between the HLA class II region and disease susceptibility. Since the trigger for podoconiosis is unknown and no animal model exists, their approach was based on characterization of PBMC's as well as RNA transcripts.

The study in general is well rounded and based on the data presented the association of podoconiosis with certain biomarkers seems valid. It is a clearly written manuscript and I only have a few comments and suggestions for changes.

It is unclear to me if their cohort was asked if they wore shoes or not. It might be a silly question to ask but if the control group in contrast to the stage 2 patients did wear shoes, then the markers might not show human genetic factors influencing the outcome of barefoot exposure to volcanic soil. The majority of the study participants were farmers, so one might assume they all showed similar behavior towards wearing shoes or not.

As mentioned above in the summary it could be interesting to double check your healthy individuals against available data from other regions of the world to further dissect relevant from irrelevant RNA transcript markers perhaps.

Supplemental Figure 1:

I would suggest to color code the CD4 and CD8 population from the first row and add larger labels on the side for the respective biomarker. In addition, I would add the controls for each CD4 and CD8 population. I can imagine how they look like, but you have space in the supplement and I think it is good practice to show the complete flow analysis and that includes the controls.

"The average percentage of CD4 and CD8 T cells expressing HLA-DR was significantly higher in podoconiosis patients compared to healthy controls ($p < 0.001$ with median of 10.7% vs 7.1% for HLA-DR on CD4 and 23.4% vs 15.8% for HLA-DR expression on CD8 cells respectively)."

What type of analysis? ANOVA with Tukey?

"Furthermore, we confirmed the increase in activation marker by comparing median MFI of each activation marker among markers ungated."

I would think the geometric mean was used? How does the frequency distribution of MFI's look like? Are they skewed towards one end or are they bimodal within a patient population? Depending on that I would consider using the geometric mean instead, otherwise a single high or low value might skew the median as they are equally weighted in the median calculation.

Figure 2 change "monos" to monocytes as it is lab-jargon

I believe Figure 5 C&D need to be swapped or the legend changed. Figure 5C shows the upregulated genes and 5D the downregulated ones. But the individual figures do not line up with the table of GO genes.

"Consistent with this possibility, we observed higher levels of TNF- α and IL-1 β in unstimulated culture wells and IL-1 β mRNA in the peripheral blood of podoconiosis patients (unpublished data)."
I think this data would be a valuable addition to include to support your manuscript, even if it is only in the supplemental material.

The shown protein networks look reasonable and are supported by experimental data from other publications. However, depending which single gene is looked up a different network of connections can be made. How did the authors decide which interactions to show?

As an example, searching only for MSR1 using the default values in the STRING database results in a partial overlap with interactions shown in Figure 5C.

Reviewer #2 (Attachment):

Negash et al. present single cell-based data for individuals with podoconiosis, which is an understudied preventable disease associated with exposure to minerals found in volcanic soils. Silica particles penetrating the skin trigger an inflammatory response and the characteristic swelling in the foot. The authors use healthy endemic individuals as their control group, which is a plus to reduce other potential environmental factors contributed by people living elsewhere in the world. If they had used instead a European or American control group significant differences might have been identified that were unrelated to podoconiosis. However, one could use publicly available transcriptomics data e.g. from the genome-based tissue expression consortium (PMID 25809799) to cross-check the control group. Elevated transcripts in the control group that also show differences in the podoconiosis population are likely unrelated to the disease. A quick check would be to verify that their biomarkers identified to be associated with podoconiosis hold true when testing the healthy population against a previously published reference transcriptome of other healthy individuals not living in the endemic area.

Using peripheral blood the authors identify several markers allowing them to categorize and distinguish healthy from early and late stage podoconiosis patients. Previous genome-wide studies in Ethiopia showed association between the HLA class II region and disease susceptibility. Since the trigger for podoconiosis is unknown and no animal model exists, their approach was based on characterization of PBMC's as well as RNA transcripts.

The study in general is well rounded and based on the data presented the association of podoconiosis with certain biomarkers seems valid. It is a clearly written manuscript and I only have a few comments and suggestions for changes.

It is unclear to me if their cohort was asked if they wore shoes or not. It might be a silly question to ask but if the control group in contrast to the stage 2 patients did wear shoes, then the markers might not show human genetic factors influencing the outcome of barefoot exposure to volcanic soil. The majority of the study participants were farmers, so one might assume they all showed similar behavior towards wearing shoes or not.

As mentioned above in the summary it could be interesting to double check your healthy individuals against available data from other regions of the world to further dissect relevant from irrelevant RNA transcript markers perhaps.

Supplemental Figure 1:

I would suggest to color code the CD4 and CD8 population from the first row and add larger labels on the side for the respective biomarker. In addition, I would add the controls for each CD4 and CD8 population. I can imagine how they look like, but you have space in the supplement and I think it is good practice to show the complete flow analysis and that includes the controls.

“The average percentage of CD4 and CD8 T cells expressing HLA-DR was significantly higher in podoconiosis patients compared to healthy controls ($p < 0.001$ with median of 10.7% vs 7.1% for HLA-DR on CD4 and 23.4% vs 15.8% for HLA-DR expression on CD8 cells respectively).”

What type of analysis? ANOVA with Tukey?

“Furthermore, we confirmed the increase in activation marker by comparing median MFI of each activation marker among markers ungated.”

I would think the geometric mean was used? How does the frequency distribution of MFI’s look like? Are they skewed towards one end or are they bimodal within a patient population? Depending on that I would consider using the geometric mean instead, otherwise a single high or low value might skew the median as they are equally weighted in the median calculation.

Figure 2 change “monos” to monocytes as it is lab-jargon

I believe Figure 5 C&D need to be swapped or the legend changed. Figure 5C shows the upregulated genes and 5D the downregulated ones. But the individual figures do not line up with the table of GO genes.

“Consistent with this possibility, we observed higher levels of TNF- α and IL-1 β in unstimulated culture wells and IL-1 β mRNA in the peripheral blood of podoconiosis patients (unpublished data).” I think this data would be a valuable addition to include to support your manuscript, even if it is only in the supplemental material.

The shown protein networks look reasonable and are supported by experimental data from other publications. However, depending which single gene is looked up a different network of connections can be made. How did the authors decide which interactions to show?

As an example, searching only for MSR1 using the default values in the STRING database results in a partial overlap with interactions shown in Figure 5C.

Response to reviewers' comments

Reviewer #1

Negash et al. characterised activation markers of monocyte and T and dendritic cell subsets from individuals that suffer from lymphedema caused by contact to irritant volcanic soil (podoconiosis). This study was based on genome-wide studies showing that disease susceptibility is associated with variation in HLA class II genes. Consequently, the authors immunophenotyped immune cells (e.g., HLA-DR, CD38, CD62L and CD40, CD86, CD36, respectively) as well as performed RNA sequencing gene expression analysis and revealed that receptors and activation markers involved in antigen processing and presentation and inflammation are increased in podoconiosis patients in comparison to healthy controls. I congratulate the authors for the very interesting results which are of great interest for researchers dealing with this disease, especially the pathogenesis of this stigmatizing disease remains unknown. However, some issues need to be resolved before I can endorse publication.

1) In total the authors analysed 64 podoconiosis and 49 healthy individuals, but analysed only 56 vs 44 (T cells), 43 vs 34 (monocytes/dendritic cells) and 21 vs 23 (transcriptomic). Can the authors explain the selection (maybe present a flow chart). Are the results consistent when only the 21 vs 23 samples analysed in the flow cytometry approach?

- A flow chart showing the sample numbers for the different tests and analysis as well as a reason for exclusion is now included in the supplementary figure 1. The sample size for the transcriptomics work was limited due to quality control issues with respect to the RNA, and also the need to match the samples on additional parameters due to the quantitative nature of the work – measuring relative up and down regulation of a large number of genes – e.g. the starting number of cells per volume, and hence RNA extracted needed to be matched. There were also technical/cost effectiveness considerations that limited our studies to batches of 24 samples (we did not have sufficient suitable samples to do 36 or 48 from each group). We did not separately analyse the flow cytometry data for this subset as it would not have enough statistical power to ensure meaningful interpretation.

2) In regards to 1), in my opinion it would be necessary to show an overview table about the study characteristics including gender, age and lymphedema stage (do the authors have information about ADLA?). Were all results controlled for a potential bias of age and gender?

- We have now included a supplementary table 1 showing the sociodemographic characteristics of study subjects including age, stage of the disease and duration of the disease. Study subjects were matched as much as possible based on gender and age, in particular for those with small sample size like the transcriptomics analysis, they were controlled for such effects during the analysis (supplementary table 3). The query about ADLA is addressed in point no 7 below, but it should be noted that acute illness on the day of sample collection, including ADLA as well as viral respiratory infections, fever, was an exclusion criterion.

3) In regards to the gating strategies, did the authors include a live and dead staining, which is very important when activation markers are analysed.

- It is true activation markers like HLA-DR are usually not straightforward for gating. Hence, we used FMO and unstained controls for gating which we have now edited on supplementary figure 2. Live-dead staining is really relevant when assessing antigen specific T cells because the responding frequency is so low that even a small amount of dead cells staining non-specifically positive can greatly impact results. However, the frequencies of activated cells measured here is much higher than typical antigen specific assays because we were not doing an in vitro stimulation with antigens. Even more importantly, the studies were done on fresh cells which would be expected to have only trace amounts of dead cells. Hence any contribution of dead cells was very unlikely to impact our results.

4) Figure 1: Were there any differences in the CD3 and/or CD4 cell number/frequencies per se? The authors mentioned that also MFI show the same pattern as the frequencies; is this also true for the cell numbers (this would prove that the findings are consistent and solid).

- We used the MFI to compare the expression of the activation markers among CD4 and CD8 population to corroborate our median proportion data but from % positive staining of CD3 and CD4 in the lymphocyte gate, there do not appear to be significant differences between the study groups.

5) Figure 2/Supplementary Figure 2: Since the authors state in the main figure classical, non- classical and intermediate monocytes these phrases should be linked to the gating strategy. The gating strategy for the intermediate monocytes (CD16+CD14+) is missing.

- The gating strategy for the intermediate monocyte is now included in supplementary figure 3

6) Can the authors explain the choice of the different activation markers? Especially, why CD80 was not analysed, which have been found upregulated in the transcriptomics analysis and is important for T cell activation.

- It is true CD80 is one of the co-stimulatory and the activation marker but we did the transcriptomics assay after the peripheral blood immunophenotyping assay. Hence, we didn't have prior information on the upregulated genes. Moreover, our flow cytometry analyser FACS Canto II measures only 8 colours so we had to accommodate all representative activation markers including CD36, CD40, HLA-DR on top of the lineage or sub population markers. If it was not for this limitation and the cost of doing multiple panels, we could have incorporated additional markers including CD80 and other markers such as the early activation marker CD69.

7) The authors state that HLA expression/activation is associated with immunity against pathogens and nicely described the associations with silica crystal uptake in regards to the pathogenesis of the disease, but as mentioned above it has been shown (e.g., in lymphatic filariasis lymphedema patients) that bacteria can drive acute dermatolymphangioadenitis attacks (ADLA) promoting the progression of the lymphedema. Recently, two publications analysed the skin microbiome of podoconiosis patient and associated distinct bacteria spp. and change of normal skin flora to high lymphedema stage legs. Thus, involvement of

bacteria/pathogens in the progression of the disease should be discussed as well, especially it nicely fits to the obtained activation patterns of the different immune cells. Maybe it is a combination of silica uptake which induces inflammation and destruction of the skin barrier allowing the entrance of pathogens driving ADLA and consequently disease progression.

- We have now added a statement in the discussion suggesting changes in skin microbiome could contribute in disease progression in podoconiosis and for the up regulated expression of markers we found in the current study.
- ADLA occurs only in the context of established lymphoedema in podoconiosis, as a clinical complication of the lymphedema which leads to skin changes and the loss of the skin's barrier function in innate immunity. So, whilst ADLA can trigger clinical deterioration in podoconiosis patients, it is unlikely to be a driver of the primary pathology. However, as suggested, it is still possible that sub-clinical infection plays a role in the aetiology of podoconiosis and we will be investigating this in future studies focusing more on the analysis of skin biopsy samples and the microbiome of podoconiosis patients, comparing stable groups with those who suffer from ADLA attacks. However, during the study reported here we enrolled only those patients who did not have signs of intercurrent infection on the day of enrolment (including ADLA), were stable and could walk to come to the nearby health facility for sample collection.

8) Line 560: should be supplementary table 2 and 3

- Supplementary table 2 and 3 are now edited 'Supplementary tables 3 and 5' on page 20, line 582

Reviewer #2

Negash et al. present single cell-based data for individuals with podoconiosis, which is an understudied preventable disease associated with exposure to minerals found in volcanic soils. Silica particles penetrating the skin trigger an inflammatory response and the characteristic swelling in the foot. The authors use healthy endemic individuals as their control group, which is a plus to reduce other potential environmental factors contributed by people living elsewhere in the world. If they had used instead a European or American control group significant differences might have been identified that were unrelated to podoconiosis. However, one could use publicly available transcriptomics data e.g. from the genome-based tissue expression consortium (PMID 25809799) to cross-check the control group. Elevated transcripts in the control group that also show differences in the podoconiosis population are likely unrelated to the disease. A quick check would be to verify that their biomarkers identified to be associated with podoconiosis hold true when testing the healthy population against a previously published reference transcriptome of other healthy individuals not living in the endemic area.

Using peripheral blood the authors identify several markers allowing them to categorize and distinguish healthy from early and late stage podoconiosis patients. Previous genome-wide studies in Ethiopia showed association between the HLA class II region and disease susceptibility. Since the trigger for podoconiosis is unknown and no animal model exists, their approach was based on characterization of PBMC's as well as RNA transcripts.

The study in general is well rounded and based on the data presented the association of podoconiosis with certain biomarkers seems valid. It is a clearly written manuscript and I only have a few comments and suggestions for changes.

It is unclear to me if their cohort was asked if they wore shoes or not. It might be a silly question to ask but if the control group in contrast to the stage 2 patients did wear shoes, then the markers might not show human genetic factors influencing the outcome of barefoot exposure to volcanic soil. The majority of the study participants were farmers, so one might assume they all showed similar behavior towards wearing shoes or not.

- You are right, we have carefully enrolled controls who have lived and farmed in the same area as the patients for at least for 10 years and who did not wear shoe consistently thus being equally exposed to the soil without developing podoconiosis. We enrolled mostly patients' unrelated neighbours, and this is already described in 'study population and design' section, page 18.

As mentioned above in the summary it could be interesting to double check your healthy individuals against available data from other regions of the world to further dissect relevant from irrelevant RNA transcript markers perhaps.

- This is an interesting suggestion but there is clear evidence that transcription levels vary from population to population, reflecting the genetic diversity across the world's different ethnic groups. Environmental factors can also affect gene transcription (e.g. exposure to other triggers of inflammations such as pathogens) and the harsh environment in areas endemic for podoconiosis will be very different to that of other groups from whom data is publicly available. For this reason, we feel the most relevant comparator population is unaffected individuals from the same genetic background who have not developed podoconiosis despite prolonged exposure to the soil trigger and are equally exposed to any other trigger of inflammation in this population. However, we appreciate the sample size is relatively small and we are addressing this by replicating the findings in the same Ethiopian population and through an on-going project in Rwandan podoconiosis patients which will include a phenotypic, GWAS and RNA-Seq study. We planned to do a comparative and merged analysis of the Ethiopian GWAS and RNA-Seq data which will help us identify a reproducible transcripts and biomarkers which we can take forward.

Supplemental Figure 1:

I would suggest to color code the CD4 and CD8 population from the first row and add larger labels on the side for the respective biomarker. In addition, I would add the controls for each CD4 and CD8 population. I can imagine how they look like, but you have space in the supplement and I think it is good practice to show the complete flow analysis and that includes the controls.

- We have now colour coded the CD4 and CD8 population based on your suggestions, increased the font of the labels for the biomarkers, and added the controls to both sides of the supplementary figure 2

“The average percentage of CD4 and CD8 T cells expressing HLA-DR was significantly higher in podoconiosis patients compared to healthy controls ($p < 0.001$ with median of 10.7% vs 7.1% for HLA-DR on CD4 and 23.4% vs 15.8% for HLA-DR expression on CD8 cells respectively).”

What type of analysis? ANOVA with Tukey?

- We have used the Mann Whitney U test for this analysis and this is mentioned in the manuscript in the ‘statistical analysis’ section, page 19.

“Furthermore, we confirmed the increase in activation marker by comparing median MFI of each activation marker among markers ungated.”

I would think the geometric mean was used? How does the frequency distribution of MFI's look like? Are they skewed towards one end or are they bimodal within a patient population? Depending on that I would consider using the geometric mean instead, otherwise a single

high or low value might skew the median as they are equally weighted in the median calculation.

- Our data was skewed to the left for most of the significantly different parameters, so we used the non-parametric Mann Whitney U test which is less affected by single high or low values compared to other parameters. We have also checked the comparison using geometric mean but got very similar p values. This is now elaborated in the statistical analysis sub section of the method and material section.

Figure 2 change “monos” to monocytes as it is lab-jargon

- Monos in Fig 2 is now changed to monocytes, page 6.

I believe Figure 5 C&D need to be swapped or the legend changed. Figure 5C shows the upregulated genes and 5D the downregulated ones. But the individual figures do not line up with the table of GO genes.

- Figure 5 C and D are now swapped to go along with GO categories, page 10.

“Consistent with this possibility, we observed higher levels of TNF- α and IL-1 β in unstimulated culture wells and IL-1 β mRNA in the peripheral blood of podoconiosis patients (unpublished data).” I think this data would be a valuable addition to include to support your manuscript, even if it is only in the supplemental material.

- We have included this statement because it would strengthen the argument we made about the increased activation markers. The cytokine response from the in vitro stimulation experiment is a big dataset by itself and has been written up as a separate manuscript. Including that data to the current manuscript would incur too much information to digest to an already large manuscript which has a lot of flow and transcriptomics data, and could also affect our ability to publish this separate piece of work as a full story.

The shown protein networks look reasonable and are supported by experimental data from other publications. However, depending which single gene is looked up a different network of connections can be made. How did the authors decide which interactions to show?

As an example, searching only for MSR1 using the default values in the STRING database results in a partial overlap with interactions shown in Figure 5C.

- We have selected genes from the gene ontology categories which had the highest enrichment score and significant p values to submit to STRING for the up regulated and down regulated genes separately, in our case they were GO categories of ‘cell membrane and signalling’ and ‘histones’ respectively (This is mentioned in methods in ‘construction of P-P interaction network’, page 21).

REVIEWERS' COMMENTS

Reviewer #1 (Remarks to the Author):

I thank the authors for their responses, clarifications and amendments o the manuscript. I do not have further questions and think that the manuscript is suitable for publication.

Reviewer #2 (Remarks to the Author):

All concerns and suggestions have been addressed for this reviewer. Good luck with publishing your other large dataset.